

# Bacillus amyloliquefaciens 11B91 inoculation enhances the growth of quinoa (Chenopodium quinoa Willd.) under salt stress

Jing Li[1,2,3], Xiaonong Guo[1,2,3], Deyu Cai[1,2,3], Ying Xu[1] and Yaling Wang[1]

[1] College of Life Science and Engineering, Northwest Minzu University, Lanzhou, China
[2] Key Laboratory of Biotechnology and Bioengineering of State Ethnic Affairs Commission, Biomedical Research Center, Northwest Minzu University, Lanzhou, China
[3] China-Malaysia National Joint Laboratory, Biomedical Research Center, Northwest Minzu University, Lanzhou, China

## ABSTRACT

Quinoa (*Chenopodium quinoa* Willd.) is a highly nutritious food product with a comprehensive development prospect. Here, we discussed the effect of *Bacillus amyloliquefaciens* 11B91 on the growth, development and salt tolerance (salt concentrations: 0, 150, 300 mmol·L$^{-1}$) of quinoa and highlighted a positive role for the application of plant growth-promoting rhizobacteria bacteria in quinoa. In this artical, the growth-promoting effect of *Bacillus amyloliquefaciens* 11B91 on quinoa (Longli No.1) and the changes in biomass, chlorophyll content, root activity and total phosphorus content under salt stress were measured. The results revealed that plants inoculated with 11B91 exhibited increased maximum shoot fresh weight (73.95%), root fresh weight (75.36%), root dry weight (136%), chlorophyll *a* (65.32%) contents and chlorophyll *b* (58.5%) contents, root activity (54.44%) and total phosphorus content (16.66%). Additionally, plants inoculated with 11B91 under salt stress plants showed significantly improved, fresh weight (107%), dry weight (133%), chlorophyll *a* (162%) contents and chlorophyll *b* (76.37%) contents, root activity (33.07%), and total phosphorus content (42.73%).

## INTRODUCTION

Today, $8.33 \times 10^8$ hectares of land around the world are affected by salt and alkalinity (*Wang et al., 2023*). Soil salinization is among the main types of land degradation. Soil salinization is considered a key issue that poses a significant threat to global food security (*Butcher et al., 2016*; *Sahab et al., 2021*). Salinization is a major limiting abiotic factor affecting the growth, development and yield of higher plants. Under saline-alkali stress, plants may have lower water absorption rates, severe ion toxicity and reactive oxygen species toxicity in cells. For plants, if the salinity level exceeds the species-dependent threshold, it will be harmful and reduce the yield of crops. The harm caused by salt may be reflected in the excessive accumulation of reactive oxygen species, ionic toxicity, damage to

Corresponding author
Xiaonong Guo,
gxnwww@xbmu.edu.cn

the antioxidant defense system and loss of photosynthetic function (*Nhung et al., 2019*). In addition, irrigation systems are prone to salinization in agriculture. About half of the world's existing irrigation systems are influenced by salinization (*Tomaz et al., 2020*). How to overcome soil salinization has become a focus around the world. In recent years, many scholars have actively considered ways to improve saline-alkali soil, and improving its productivity has become a new research hotspot.

Quinoa (*Chenopodium quinoa* Willd.), is in the Chenopodiaceae family, has high salt tolerance, grows to be 1–2 m tall, produces flat seed, is an annual herb and can be used as a cereal crop (*Angeli et al., 2020*; *Pathan & Siddiqui, 2022*). Quinoa is rich in various amino acids and essential vitamins, as well as proteins, flavonoids, saponins, polyphenols, and dietary fiber (*Liu et al., 2022*). Quinoa, like many common crops, contains a group of saponins belonging to the class of triterpene glycosides, including phenolic acid, eicosatriene saponins and phylogenesis (*Gómez-Pando, Aguilar-Castellanos & Ibañez-Tremolada, 2019*). These secondary metabolites are mainly found in quinoa bran and quinoa seeds (*Lin et al., 2019*) These bioactive ingredients from quinoa have anti-inflammatory, antioxidant and lipid-lowering potential (*Guo et al., 2020*; *Dong et al., 2020*). Due to the nutritional and physicochemical components of quinoa, it has potential as a functional food (*Filho et al., 2017*; *Tang et al., 2017*). Quinoa seed can be used to make flour, soup, cereals and alcohol. It can be widely used in the food, daily chemical, agricultural, and pharmaceutical industries (*Yao et al., 2019*). Studies have shown that quinoa can produce seeds even under the salinity of seawater. In the case of low salinity, the moist weight of quinoa can increase temporarily (*Katsunori et al., 2019*). Moderate application of NaCl can promote growth in quinoa, and excessive application of NaCl has an inhibitory effect (*Yang et al., 2017*).

Plant growth-promoting rhizobacteria (PGPRs) comprise a soil bacterial community that can promote plant growth, reduce the incidence of plant diseases and improve plant resistance to stress (*Agrawal & Archana, 2021*). In the past 10 years, the study of plant growth-promoting bacteria has become a hot topic throughout the world, as they have many mechanisms to promote plant growth. The action of PGPRs in plants can be classified into direct and indirect effects. The direct effects mainly include the production of plant hormones, nitrogen fixation and phosphorus dissolution as biological fertilizer to promote the absorption of plant nutrition, and the indirect effects are mainly reflected in the production of antibacterial compounds as biological protectors to inhibit the growth of plant diseases. Some studies have found that PGPRs can also reduce pesticides, overcome pesticide residues and ensure the safety and health of crops (*Gani et al., 2021*). In recent years, there has been a lot of research on PGPRs and plant stress resistance physiology, such as promoting growth, improving salt resistance, and improving drought resistance and disease resistance (*Chandran, Meena & Swapnil, 2021*). The impact of PGPR on the growth of lettuce, tomato, rice, and wheat under salinity stress was investigated, and the detrimental effects of salinity were mitigated (*González et al., 2023*; *Patani et al., 2023*; *Tiwari et al., 2023*; *Zahra et al., 2023*). The interaction between PGPRs and plants under salt stress has shown that PGPRs, which produce ACC deaminase, can induce salt tolerance in barley under salt stress, thus improving plant growth (*Azadikhah et al., 2019*).

WU-9 inoculation under salt stress mainly improves salt tolerance and plant growth by regulating salt stress-responding ethylene and auxin signal transduction, the utilization of proline, photosynthesis, antioxidant enzyme activities and cell enlargement (*Wang et al., 2022*). Therefore, it is of profound significance to screen and apply PGPR strains to improve crop growth and adaptability to adversity. Nevertheless, the interaction between PGPRs and quinoa under salt stress has rarely been reported. In this experiment, we studied the effects of plant growth-promoting rhizobacteria on the growth and salt tolerance of quinoa, providing important resource materials for the research and application of microbial remediation or microbial phytoremediation of saline alkali land.

## MATERIALS AND METHODS

### Laboratory materials

*Bacillus amyloliquefaciens* 11B91 was obtained from the Northwest Institute of Eco-Environmental and Resources, Quinoa seed (Longli No. 1) was obtained from the Animal Husbandry, Pasture and Green Agriculture Institute (Gansu, China).

### Addition of 11B91 under normal conditions and salt treatment

The seeds were disinfected by soaking quinoa seed in 0.5% $KMnO_4$ for 10 min and rinsing in sterile water until it was colorless. For the seed soaking treatment, Luria-Bertani (LB) liquid medium was first prepared by mixing of distilled water (1 L) with tryptone (10 g), yeast extract (5 g), and sodium chloride (10 g) according to the recipe at pH 7.2 ± 0.2, and then sterilizing it at 121 °C for 20 min; the LB liquid medium was inoculated with strain *Bacillus amyloliquefaciens* 11B91 and incubated at 28 °C and 180 r/min to facilitate rapid culture(the $OD_{600}$ value was approximately 1) (*Jira et al., 2018*). Quinoa seeds were soaked in LB liquid medium (blank control group) and LB liquid medium with *Bacillus amyloliquefaciens* 11B91 (treatment group) for 5 min, ensuring that the seeds were completely submerged (*He et al., 2018*).

After seed soaking, the seeds were cultured in a petri dish with sterile filter artical. After 3 days of germination, the seeds were transferred to a plate containing sterilized vermiculite and cultured with half-strength Hoagland's nutrient solution. (containing 2 mM $KNO_3$, 0.5 mM $NH_4H_2PO_4$, 0.5 mM $Ca(NO_3)_2$, 0.5 mM $MgSO_4$, 0.5 mM Fe-citrate, 92 μM $H_3BO_3$, 18 μM $MnCl_2 \cdot 4H_2O$, 1.6 μM $ZnSO_4 \cdot 7H_2O$, 0.6 μM $CuSO_4 \cdot 5H_2O$, and 0.7 μM $(NH_4)_6Mo_7O_{24} \cdot 4H_2O$) (2 L). Relevant physiological data were measured at 2, 3 and 4 weeks of quinoa growth (*Zhao et al., 2016*).

After 4 weeks of quinoa growth, salt treatments were carried out by adding a half-strength Hoagland nutrient solution with salt concentrations of 0, 150 and 300 mmol/L NaCl. Three replicates were set up for each treatment. Relevant physiological data were measured at the first, second, and third weeks of the salt treatment (*Zhao et al., 2016*).

### Determination of biomass

The seedlings of each treatment group were cleaned. An analytical balance was used to determine the fresh weight of the shoots and roots. After being weighed, the samples were

oven-dried at 60 °C for 72 h, and after cooling for 48 h, the dry weight was obtained. Each parameter was measured in six replicates.

## Determination of chlorophyll content

Fresh leaves (0.05 g) were weighed and recorded by a specific weight (G). After crushing, they were placed in 10 mL centrifuge tubes, and 10 mL ethanol acetone (1:1) mixed solution was added. Samples were stored in the light for 48 h, and the absorbance value was determined with UV-spectrophotometer (UV-1800; Shimadzu, Suzhou, China) at 663 and 645 nm wavelengths (using solution as a blank control) (*Zhang et al., 2019*). The contents of chlorophyll *a* (Chl *a*) and chlorophyll *b* (Chl *b*) were calculated as follows:

$$\text{Chl } a = \frac{(12.72A_{663} - 2.59A_{645})V}{1000W}$$

$$\text{Chl } b = \frac{(22.88A_{645} - 4.67A_{663})V}{1000W}$$

Note: $W$: veight of fresh leaves; $V$: volume of ethanol acetone (1:1) mixed solution; A663 and A645: the absorbance value was determined with a spectrophotometer at 663 and 645 nm wavelengths, respectively.

## Root activity

The triphenyl tetrazolium chloride (TTC) colorimetric method was used to determine root activity in quinoa. Quinoa roots (0.5 g) were mixed with 10 mL of 0.4% TTC solution, succinic acid and phosphoric acid buffer in equal amounts. A TTC standard curve was obtained, and the TTC reduction was calculated against the standard curve (*Muhammad et al., 2019*).

$$Root\ activity = \frac{Ch}{w}$$

Note: $C$: TTC (µg), $w$: root weight (g), $h$: time (h).

## Total phosphorus content

For the determination of the total phosphorus content, a method described by *Tisarum et al. (2020)* was used. Data were collected as previously described (*Cai et al., 2021*). Specifically, masses of dried and ground quinoa plants, including the roots, were weighed, decocted with hydrogen peroxide ($H_2O_2$) and concentrated $H_2SO_4$, allowed to cool and then fixed in ultrapure water. Then, the total phosphorus content of the quinoa was determined using the molybdenum blue colorimetric method. Total P (mg·g$^{-1}$ DW) was measured at 420 nm with a UV-spectrophotometer using potassium dihydrogen phosphate as the calibration standard.

## Statistical analyses

Excel 2010 was used to collect and process the original data. SPSS version 21.0 (IBM Corporation, Chicago, IL, USA) was used to identify differences between means at a significance level of $p \leq 0.05$. The Duncan multi-interval test was used for one-way analysis

of variance (ANOVA). The results of the growth and physiological data are presented as the mean ± standard error. The data were plotted using ORIGIN software (*Zhao et al., 2016*).

## RESULTS

### 11B91 treatment affected the biomass of quinoa seedlings under normal conditions

In the 11B91 treatment, the fresh weight of the aboveground parts at the second and third weeks after inoculation and the fresh weight of the roots at the third week were significantly higher than that in the blank group ($p < 0.05$). In the second and third weeks of the 11B91 treatment, the fresh weight of the aboveground parts was 59.54% and 73.95% higher than that of the blank group (Fig. 1A), respectively. There was no significant difference in the aboveground fresh weight of 11B91-treated seedlings in the first week. In the third week of 11B91 treatment, the fresh weight of the roots was 75.36% higher than that of the blank group (Fig. 1B). There was no significant difference in fresh root weight ($p < 0.05$) between the first and second weeks of 11B91-treated seedlings, and the fresh root weight of 11B91—treated seedlings was 49.66% higher than the blank group in the third week (Fig. 1B).

There was no significant difference in aboveground dry weight of 11B91-treated seedlings in the first, second and third weeks (Fig. 1C). In the first and second weeks, the dry weight of the 11B91-treated seedlings increased by 27.46% and 16.83%, respectively. In the second week, the dry weight of the roots of the 11B91-treated seedlings was significantly higher than that of the blank group ($p < 0.05$). The dry root weight of the 11B91-treated seedlings was 1.36 times higher than that of the blank group. In the first and third weeks, there was no significant difference in root dry weight of 11B91-treated seedlings ($p < 0.05$). At the third week of 11B91 treatment, the dry root weight was 49.56% higher than that of the blank group (Fig. 1D).

### 11B91 treatment affected the chlorophyll content of quinoa seedlings under normal conditions

The content of chlorophyll *a* in the first and third weeks of quinoa seedlings treated with *B. amyloliquefaciens* 11B91 was significantly higher than that of the blank group ($p < 0.05$) (Fig. 2A). The chlorophyll *a* content in the first and third weeks of the 11B91 treatment was 65.32% and 23.48% higher than that in the blank group. There was no significant difference in chlorophyll *a* content in the second week of 11B91-treated seedlings ($p < 0.05$), and the content of chlorophyll *a* in seedlings treated with 11B91 was 11.23% higher than in the blank group (Fig. 2A).

The content of chlorophyll *b* in the first week of quinoa seedlings treated with *B. amyloliquefaciens* 11B91 was significantly higher than that in the blank group ($p < 0.05$); the content of chlorophyll *b* in the first week of seedlings treated with 11B91 was 58.50% higher than that in the blank group (Fig. 2B). There was no significant difference in chlorophyll *b* content in the second and third weeks of 11B91-treated seedlings ($p > 0.05$);

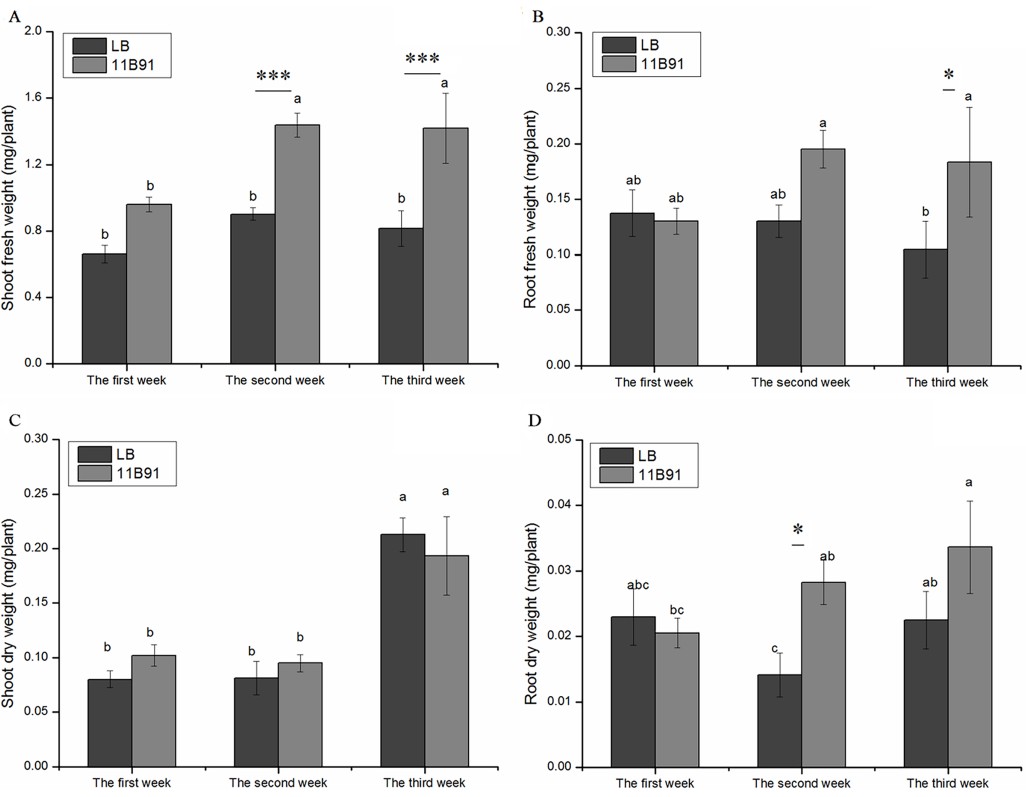

**Figure 1 Effect of 11B91 on the growth of quinoa.** Quinoa's (A) shoot fresh weight, (B) root fresh weight, (C) shoot dry weight, and (D) root dry weight. Values are presented as means, and bars indicate standard errors ($n = 6$). Columns with different letters indicate significant differences among treatments at $p < 0.05$ (ANOVA and Duncan's multiple comparison test). An asterisk (*) refers to the significant difference between the LB and 11B91 media ($p < 0.05$). Three asterisks (***) refer to the significant difference between the LB and 11B91 media ($p < 0.001$).

the content of chlorophyll *b* in 11B91-treated quinoa seedlings was 45.42% higher than that in the blank group in the third week (Fig. 2B).

## Effects of 11B91 on the root activity of quinoa seedlings under normal conditions

The root activity was significantly higher in 11B91-treated seedlings than in the blank group in the first and second weeks ($p < 0.05$; Fig. 3) and was 54.44% and 5.45% higher, respectively.

## Effects of 11B91 on the total phosphorus content of quinoa seedlings

The total phosphorus content of quinoa seedlings treated with 11B91 was significantly higher (16.66%; $p < 0.05$) than that of the blank group in the first week (Fig. 4).

## Effect of 11B91 on quinoa growth under salt treatment

Under *B. amyloliquefaciens* 11B91 treatment, in the first and second weeks of 0 mM salt treatment, in the first, second and third weeks of 150 mM salt treatment, and in the first and third weeks of 300 mM salt treatment, the fresh weight was significantly higher than

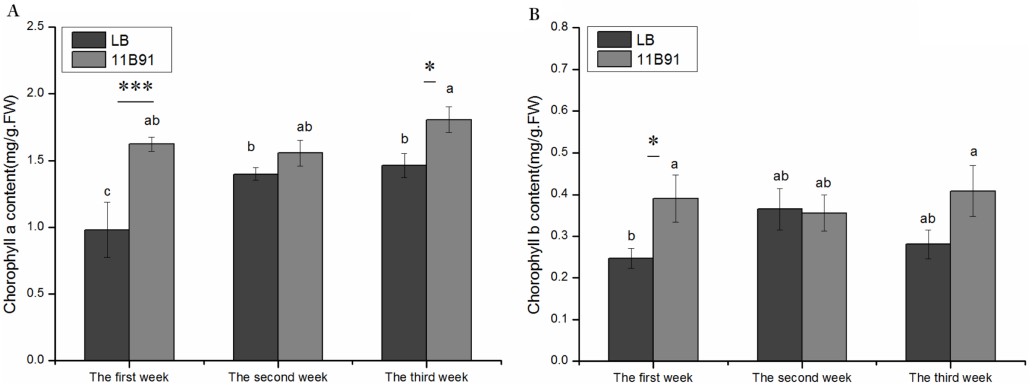

**Figure 2 Effects of 11B91 on the leaf chlorophyll a content (A) and leaf chlorophyll b content (B) of quinoa.** Values are presented as means, and bars indicate standard errors (SEs) ($n = 6$). Columns with different letters indicate significant differences among treatments at $p < 0.05$ (ANOVA and Duncan's multiple comparison test). An asterisk (*) refers to the significant difference between the LB and 11B91 media ($p < 0.05$). Three asterisks (***) refer to the significant difference between the LB and 11B91 media ($p < 0.001$).

that without 11B91 treatment ($p < 0.05$). The fresh weight of quinoa treated with 11B91 was 63.99% and 83.55% higher than that of the blank group in the first and second weeks of 0 mM salt treatment (Figs. 5A and 5C). The fresh weight in the 150 mM salt treatment was 57.59%, 88.18% and 42.89% higher than that in the blank group at 1, 2 and 3 weeks, respectively (Figs. 5A, 5C and 5E). Under 300 mM salt treatment, the fresh weight was 66.98% and 1.07 times higher than that of the blank group in the first and third weeks (Figs. 5A, 5E). There was no significant difference in the fresh weight between the blank group and 11B91 treatment in the 0 mM salt treatment in the third week and the second week under 300 mM ($p < 0.05$). The fresh weight in the 11B91 treatment was 23.98% higher than that of the blank group in the third week of 0 mM salt treatment and 54.99% higher in the second week of 300 mM salt treatment than in the blank group (Figs. 5C and 5E).

The dry weight of 11B91-treated quinoa seedlings treated with 150 mM salt in the second week and 300 mM salt in the first week was significantly higher than that of the blank group ($p < 0.05$) (Figs. 5B and 5D). The dry weight of the 150 mM salt treatment was 64.86% higher than that of the blank group in the second week of 11B91 treatment; the dry weight was 1.33 times, 75.99% and 45.03% higher than in the blank group in the 300 mM salt treatment. Under the *B. amyloliquefaciens* 11B91 treatment, in the first, second and third weeks of 0 mM salt treatment, in the first and third weeks of 150 mM salt treatment, and in the second and third weeks of 300 mM salt treatment, the dry weight showed no significant difference compared to the treatment without 11B91 ($p < 0.05$). No significant difference was found in dry weight in the 0 mM treatment at three weeks or the treatment during the first and third weeks of salt treatment.

The dry weight in the 11B91 treatment was 21.07% and 11.26% higher than that of the blank group in the 0 mM salt treatment in the first and third weeks, respectively. In the 150 mM salt treatment in the first and third weeks, the dry weight was 20.54% and 10.85% higher than that in the blank group (Figs. 5B and 5F).

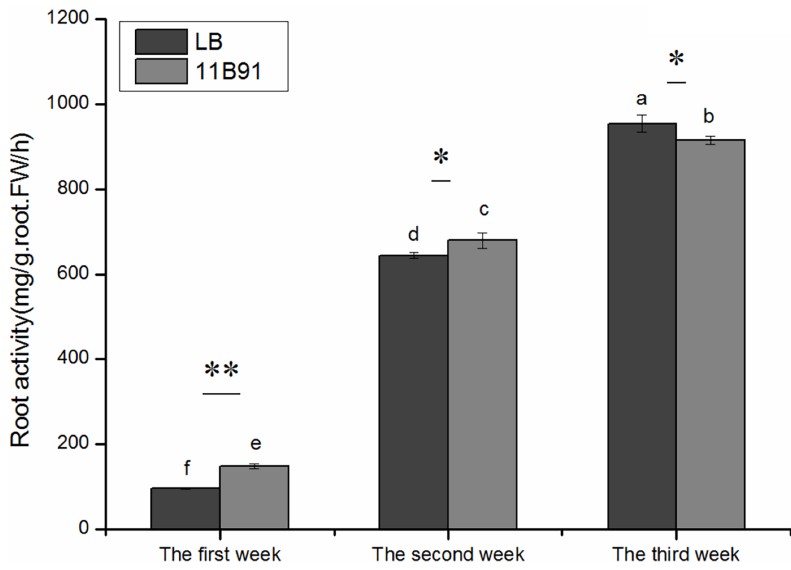

**Figure 3 Effects of 11B91 on the root activity of quinoa.** The values are presented as means, and the bars indicate the SE ($n = 3$). The columns with different letters show the significant differences among treatments at $p < 0.05$ (ANOVA and Duncan's multiple comparison test). An asterisk (*) refers to the significant difference between the LB and 11B91 mediums ($p < 0.05$). Two asterisks (**) refer to the significant difference between the LB and 11B91 mediums ($p < 0.01$).

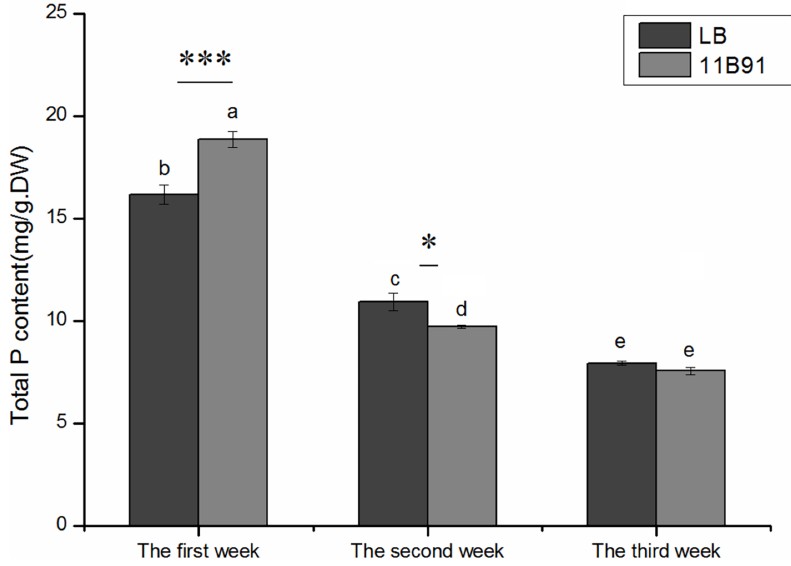

**Figure 4 Effects of 11B91 on total phosphorus content of quinoa seedlings.** Values are presented as means, and bars indicate SEs ($n = 6$). Columns with different letters indicate significant differences among treatments at $p < 0.05$ (ANOVA and Duncan's multiple comparison test). An asterisk (*) refers to the significant difference between the LB and 11B91 media ($p < 0.05$). Three asterisks (***) refer to the significant difference between the LB and 11B91 media ($p < 0.001$).

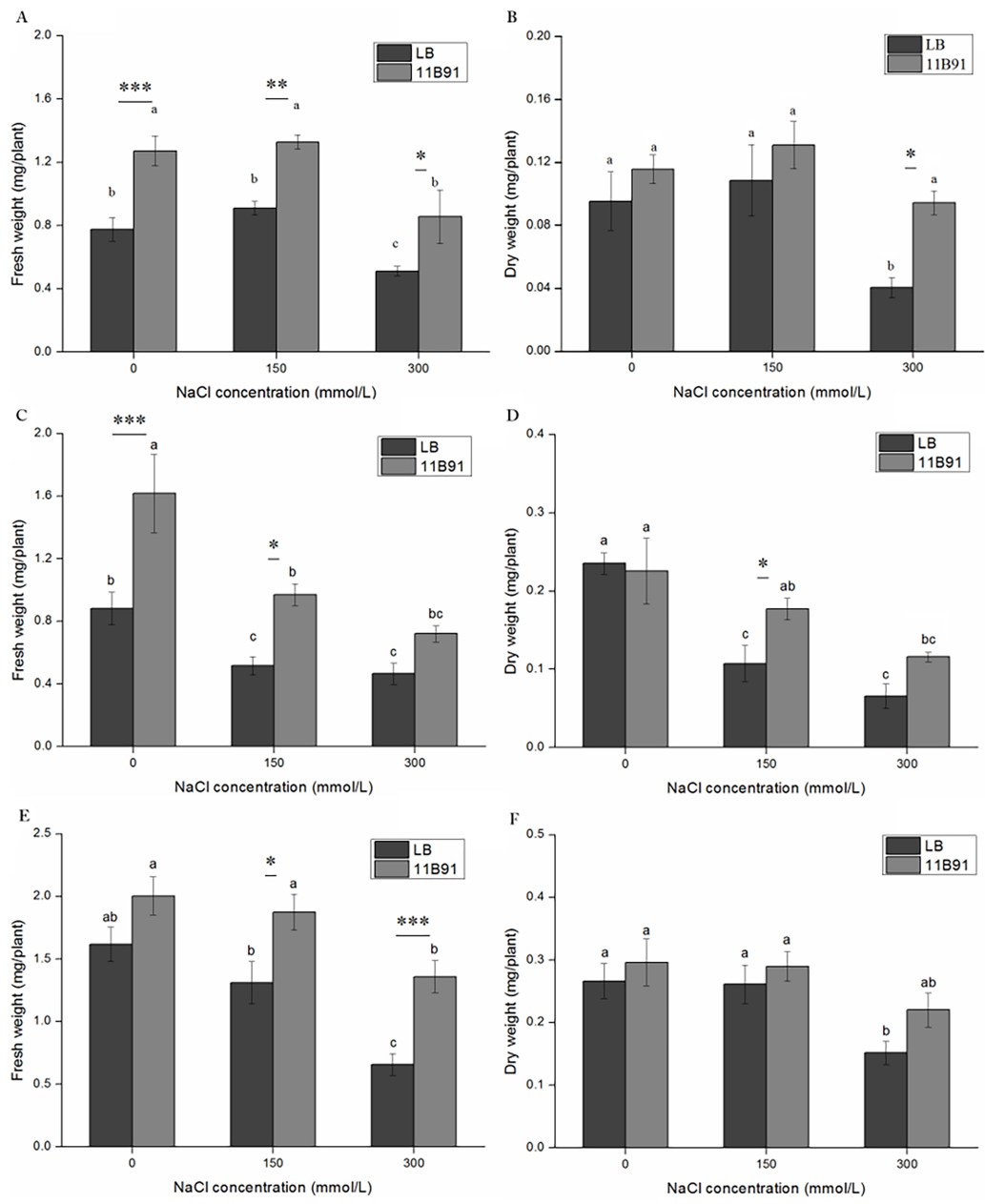

**Figure 5** **Effects of 11B91 on fresh weight and dry weight of quinoa under salt treatment (0, 150, and 300 mM of NaCl).** Values are presented as means, and bars indicate SEs ($n = 3$). Columns with different letters indicate significant differences among treatments at $p < 0.05$ (ANOVA and Duncan's multiple comparison test). (A and B) NaCl treatment for 7 d; (C and D) NaCl treatment for 14 d; (E and F) NaCl treatment for 21 d. An asterisk (*) refers to the significant difference between the LB and 11B91 media ($p < 0.05$). Two asterisks (**) refer to the significant difference between the LB and 11B91 media ($p < 0.01$). Three asterisks (***) refer to the significant difference between the LB and 11B91 media ($p < 0.001$).

## Effect of 11B91 on the chlorophyll content of quinoa under salt treatment

The content of chlorophyll *a* in quinoa treated with 11B91 was significantly higher than that in the blank group ($p < 0.05$) in the 0 mM salt treatment at 2 and 3 weeks, 150 mM salt treatment in the first week, and 300 mM salt treatment in the first, second and third weeks (Figs. 5A, 6C and 6E). The chlorophyll *a* content in the second and third weeks in the 0 mM salt treatment was 23.48% and 57.79% higher than that in the blank group, respectively. The chlorophyll *a* content in the first week with 150 mM salt treatment was 91.36% higher than that in the blank group, and under 300 mM salt treatment, it was 95.07%, 162 % and 34.76% higher than that in the blank group at 1 and 2 weeks, respectively. No significant difference was found in the chlorophyll *a* content between seedlings in the 0 mM salt treatment in the first week, 150 mM salt treatment in weeks 2 and 3($p < 0.05$). The chlorophyll *a* content in the first week with 11B91 in the 0 mM salt treatment was 11.23% higher than that in the blank group (Figs. 6A and 6E).

The chlorophyll *b* content in the second week of 0 mM salt treatment and in the second week of 300 mM salt treatment was significantly higher than that in the blank group ($p < 0.05$) (Fig. 6D). The chlorophyll *b* content in the second week of 11B91 and 0 mM salt treatment was 45.42% higher than that in the blank group, and in the 300 mM salt treatment, it was 76.37% higher in the second week than in the blank group.

The chlorophyll *b* content between seedlings in the 0 mM salt treatment in the first and third weeks, 150 mM salt treatment in the first, second and third weeks, and 300 mM salt treatment in the first and third weeks was not significantly different ($p < 0.05$).

The chlorophyll *b* content in the third week of the 0 mM salt treatment was 24.82% higher than that in the blank group, and in the first and third weeks of the 300 mM salt treatment, it was 109% and 20.53% higher than that in the blank group, respectively (Fig. 6F).

## Effect of 11B91 on quinoa root activity under salt treatment

Under 11B91 treatment, in the first week with 150 mM salt treatment the root activity was significantly lower than in the blank group. Under 11B91 treatment, in the second and third weeks with 150 mM salt treatment and in the third week with 300 mM salt treatment, the root activity was significantly higher than in the blank group (Figs. 7A–7C). The root activity under 11B91 treatment in the second and third weeks of 150 mM salt treatment was 7.17% and 33.07% higher than that in the blank group, respectively, and 26.87% higher in the third week after 300 mM salt treatment than in the blank group (Fig. 7C). There was no significant difference in root activity between the 11B91 and blank groups in the first, second and third weeks of 0 mM salt treatment and in the first and second weeks of 300 mM salt treatment ($p < 0.05$). The root activity of the 11B91-treated seedlings was 5.45% higher than that of the blank group in the first week of 0 mM salt treatment (Fig. 7A).

## Effect of 11B91 on the total P content of quinoa under salt treatment

The total phosphorus content of plants in all LB treatments showed a decreasing trend with increasing salt concentrations at different salt treatment times (Figs. 8A–8C). The inoculation of 11B91 maintained a relatively stable P content in plants under salt

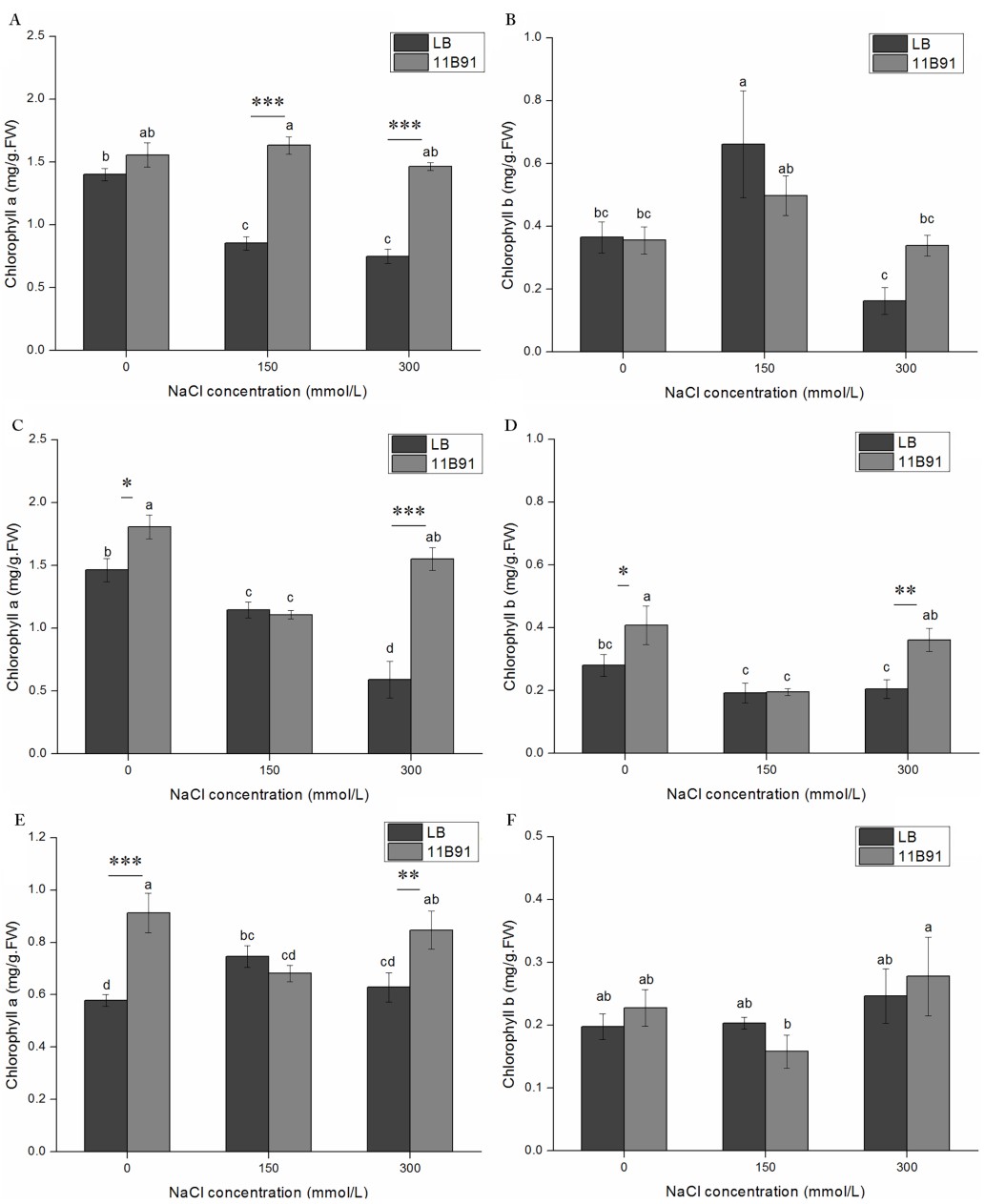

**Figure 6 Effects of 11B91 on the leaf chlorophyll a content and leaf chlorophyll b content of quinoa under salt stress (0, 150, and 300 mM of NaCl).** Values are presented as means, and bars indicate standard errors (SEs) ($n = 6$). (A and B) NaCl treatment for 7 d. (C and D) NaCl treatment for 14 d. (E and F) NaCl treatment for 21 d. Columns with different letters indicate significant differences among treatments at $p < 0.05$ (ANOVA and Duncan's multiple comparison test). An asterisk (*) refers to the significant difference between the LB and 11B91 media ($p < 0.05$). Two asterisks (**) refer to the significant difference between the LB and 11B91 media ($p < 0.01$). Three asterisks (***) refer to the significant difference between the LB and 11B91 media ($p < 0.001$).

stress. In the second week, the total phosphorus content under 150 and 300 mM salt treatment in seedlings treated with 11B91 was significantly higher than that of the blank group ($p < 0.05$). In the second week, the total phosphorus content of the 11B91 treatment

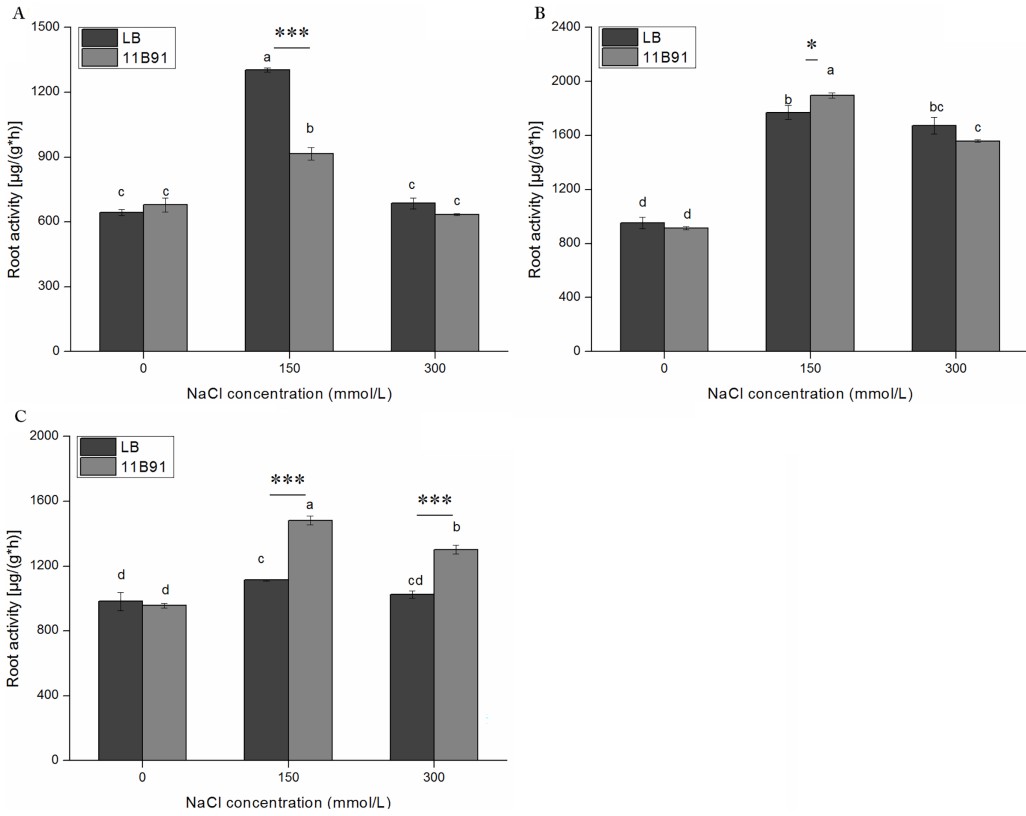

**Figure 7 Effects of 11B91 on root activity of quinoa seedlings under salt treatment.** The root activity content of plants measured after (A) 7 d, (B) 14 d, and (C) 21 d. Values are presented as means, and bars indicate the SEs ($n = 6$). Columns with different letters indicate significant differences among treatments at $p < 0.05$ (ANOVA and Duncan's multiple comparison test). An asterisk (*) refers to the significant difference between the LB and 11B91 media ($p < 0.05$). Three asterisks (***) refer to the significant difference between the LB and 11B91 media ($p < 0.001$).

was 7.40% higher under 150 mM salt treatment than that in the blank group and 42.73% higher under 300 mM salt treatment than that in the blank group (Fig. 8B). There was no significant difference in total phosphorus content between the 11B91 and blank groups in the first and third weeks of 150 and 300 mM salt treatment (Figs. 8A and 8C).

## DISCUSSION

Mechanisms by which PGPRs promote plant growth include the ability to produce or alter phytohormone concentrations, induce systemic resistance in crops, and dissolve mineral phosphorus and other nutrients to promote plant growth. In this experiment, physiological indicators, such as the dry and fresh weight of shoots and roots of quinoa, were determined. The experimental results showed that the fresh weight of quinoa shoots after 11B91 treatment was higher than that of those without 11B91 treatment, and the growth-promoting effect became more obvious over time. There was no significant increase in the shoot dry weight of quinoa treated with or without 11B91. Our results are in agreement with those of previous studies, which showed that two PGPR strains selected from perennial ryegrass promoted the dry and fresh weights of shoots and roots (*Niu,*

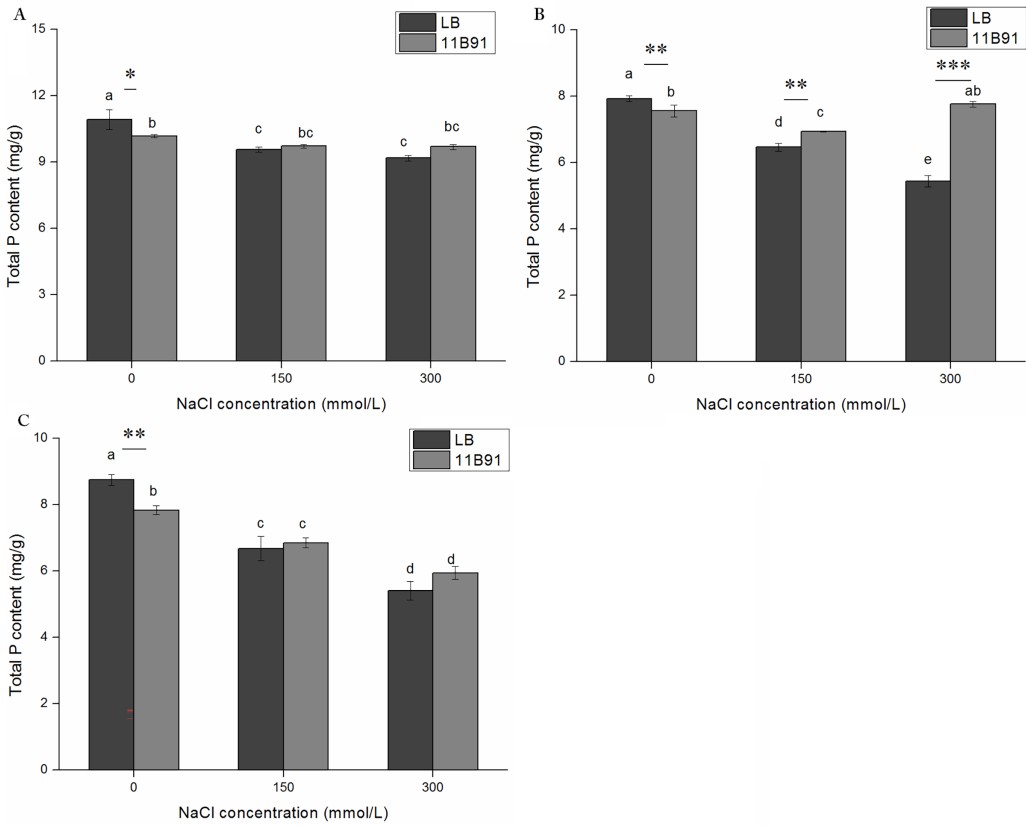

**Figure 8 Effects of 11B91 on total phosphorus content of quinoa seedlings under salt treatment.** The total P content of plants measured after (A) 7 d, (B) 14 d, and (C) 21 d. Values are presented as means, and bars indicate the SEs ($n$ = 6). Columns with different letters indicate significant differences among treatments at $p < 0.05$ (ANOVA and Duncan's multiple comparison test). An asterisk (*) refers to the significant difference between the LB and 11B91 media ($p < 0.05$). Two asterisks (**) refer to the significant difference between the LB and 11B91 media ($p < 0.01$). Three asterisks (***) refer to the significant difference between the LB and 11B91 media ($p < 0.001$).

*2017*) and that rhizosphere growth-promoting bacteria isolated from highland barley significantly promoted the germination of highland barley seeds (*Zhao et al., 2018*).

Chlorophyll is the main pigment of plant photosynthesis, and its content is an important index for reflecting photosynthesis. The PGPR microbial inoculants increased the chlorophyll content of *A. sativa*, *M. sativa*, and *C. sativus* seedlings (*Li et al., 2020*). The content of chlorophyll *a* and b in quinoa seedlings inoculated with 11B91 was higher than that of quinoa seedlings inoculated with the LB blank control group at the first and third weeks, showing a trend of decreasing first and then increasing. This statement can be supported by *Chebotar et al. (2022)* who showed that the inoculation of PGPR *Bacillus velezensis* BS89 led to an increase in the chlorophyll content of strawberry plants compared to the uninoculated control.

The root is an organ that absorbs water and nutrients and plays an important role in the growth and development of plants. The PGPR microbial inoculants increased the dry weight, plant height, root length, root average diameter, root surface area, and root volume

of *A. sativa*, *M. sativa*, and *C. sativus* seedlings (*Li et al., 2020*). Similarly, in this experiment, the root activity of quinoa was determined. The results showed that the root activity of the seedlings inoculated with 11B91 promoted by the rhizosphere was higher than that of the LB sterile control group in the first and second weeks. The total phosphorus content of quinoa seedlings inoculated with 11B91 was significantly higher than that of the LB group in the first week ($p < 0.05$). However, the root activity was higher in the control group than in the group treated with 11B91 in the third week. The total phosphorus content of quinoa seedlings inoculated with 11B91 began to decline after the second week. This is because *Bacillus* can quickly occupy a dominant position in the root, but over time, the relative abundance gradually decreases, while strains of the genus *Myxococcus* continue to exist and dominate in the root (*Fracchia et al., 2021*; *Lyu, Rachel & Donald, 2022*).

Low salt stress slows the growth rate of plants, and high salt stress causes the growth and development of plants to be in a state of obvious stagnation. Inoculation of rhizosphere growth-promoting bacteria *Pseudomonas sp.*, *Bacillus sp.* and *Enterobacter sp* with different salt concentrations can promote the germination of *inebrians* seeds and promotes the growth of rice and millet (*Ju et al., 2023*; *Sagar et al., 2020*). In this artical, physiological indexes, such as the dry and fresh weight of quinoa under salt treatment, were determined experimentally. The experimental results showed that 11B91—treated quinoa increased the total fresh weight and dry weight of 300 mM salt treatment compared with non-11B91-treated quinoa, and the effect increased with the concentration of salt treatment. The total dry weight of quinoa seeds inoculated with 11B91 and without 11B91 at 0 and 150 mM of salt treatment did not increase significantly. The probable reason is that quinoa is a halophyte with salt tolerance and a certain tolerance to the treatment concentration (*Mahdi et al., 2022*).

Chlorophyll content is an important indicator of plant photosynthesis. However, the chlorophyll content will still decrease under salt, drought and other abiotic stressors. Studies have demonstrated that the chlorophyll content of plants with weak stress resistance decreases faster than that of plants with strong stress resistance (*Li, Song & Sun, 2023*). Studies have shown that salt-tolerant growth-promoting bacteria isolated from the rhizosphere soil of plants in saline-alkali areas can increase the chlorophyll content of *Medicago sativa* L. under salt stress (*Miao et al., 2022*). The chlorophyll content of quinoa under salt treatment was determined in this experiment. The chlorophyll *a* content of quinoa seedlings inoculated with 11B91 was significantly increased in the second and third weeks of 0 mM salt treatment, the second week of 150 mM salt treatment, and the first and second weeks of 300 mM salt treatment. The chlorophyll *b* content was significantly increased in the second week of 0 mM, and in the first and second weeks of 300 mM salt treatment, it was higher than that of the LB blank control group. The same results were noted in white clover and ryegrass (*Diagne et al., 2020*; *Sapre, Gontia-Mishra & Tiwari, 2021*).

Phosphorus is associated with the formation of biomacromolecules and compounds in plants, photosynthesis, and stress resistance. Phosphorus is an indispensable element for plant growth and development. Studies have shown that some rhizosphere

growth-promoting bacteria play a promoting role in phosphorus uptake by plants. PGPR bacterial fertilizer alone or in combination significantly increases the total phosphorus content of oats (*Olasupo et al., 2022*). The root activity and total phosphorus content of quinoa were determined under salt treatment. The vitality of seedlings inoculated with 11B91 and treated with 150 mM salt for the first and second weeks and treated with 300 mM for the third week was significantly higher than that of the control group. Under 11B91 treatment, the total phosphorus content in the first week of 150 mM salt treatment and the second and third weeks of 300 mM salt treatment was higher than that of the LB blank control group. These results are consistent with previous reports which showed increased root activity and total phosphorus content in plants inoculated with salt-tolerant PGPR under salt stress (*Zilaie et al., 2022*; *Huang et al., 2022*).

## CONCLUSIONS

Taken together, the findings of this study suggest that inoculation with *Bacillus amyloliquefaciens* 11B91 could be suitable for promoting plant growth in quinoa. Under salt stress, 11B91 promoted the growth of quinoa and increased the root and total biomass of quinoa seedlings. The chlorophyll *a* and chlorophyll *b* content in the 300 mM NaCl treatment was increased by 11B91 inoculation. 11B91 improved the root activity of seedlings under salinity in the third week; it also improved the total phosphorus content of seedlings under salinity. These data provide evidence that 11B91 inoculation can improve the salt tolerance of quinoa seedlings.

## ACKNOWLEDGEMENTS

The authors acknowledge Dr. Wang Ruoyu for providing experimental materials.
We thank LetPub for its linguistic assistance during the preparation of this manuscript.

### Funding

This work was supported by the Fundamental Research Funds for the Central Universities (grant number 31920220136). The funders had no role in study design, data collection and analysis, decision to publish, or preparation of the manuscript.

### Grant Disclosures

The following grant information was disclosed by the authors:
Fundamental Research Funds for the Central Universities: 31920220136.

### Competing Interests

The authors declare that they have no competing interests.

### Author Contributions

- Jing Li conceived and designed the experiments, performed the experiments, analyzed the data, prepared figures and/or tables, authored or reviewed drafts of the article, and approved the final draft.

- Xiaonong Guo conceived and designed the experiments, authored or reviewed drafts of the article, and approved the final draft.
- Deyu Cai conceived and designed the experiments, performed the experiments, analyzed the data, prepared figures and/or tables, and approved the final draft.
- Ying Xu conceived and designed the experiments, performed the experiments, prepared figures and/or tables, and approved the final draft.
- Yaling Wang conceived and designed the experiments, authored or reviewed drafts of the article, and approved the final draft.

## Data Availability

The raw measurements are available in the Supplemental File.

## Supplemental Information

Supplemental information for this article can be found online at http://dx.doi.org/10.7717/peerj.15925#supplemental-information.

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
