# Peer review of "Bacillus amyloliquefaciens 11B91 inoculation enhances the growth of quinoa (Chenopodium quinoa Willd.) under salt stress"

_PeerJ, doi:10.7717/peerj.15925_

## Round 0.1 · original submission · Major Revisions

Please cite more recent references 2019-2023

Reviewer 2 has requested that you cite specific references. You may add them if you believe they are especially relevant. However, I do not expect you to include these citations, and if you do not include them, this will not influence my decision.

Reviewer 1 ·

Basic reporting

This study investigated the effects of the bacterium bacillus on the growth of quinoa seedlings under normal and salt stress conditions. The study was well-designed, but the writing of the manuscript could be improved, particularly in terms of the completeness of the materials and methods section, the precision of the results and conclusions, and the discussion section. Please see my detailed comments below.

Experimental design

1. Line 16: Can authors please find some updated reference to support your statement ‘Today’? Data from 30 years ago cannot represent today.
2. Lines 57-58: Do authors mean reducing the application of pesticides?
3. Line 76: What is the cultivar or variety of the quinoa seeds in this study?
4. Line 79: Please include the full name of LB liquid medium.
5. Line 91: What is the temperature setting in the oven?
6. Line 99-100: Please explain the abbreviations and letters used in the equation.

Validity of the findings

7. In figure 1D, according to the letters above the bar, there was a significant difference between bacillus treatment and the control on the second week, but it was not marked by asterisk.
8. Figure 2 title: There are both chlorophyll a and b, authors may delete ‘a’.
9. In materials and methods, there is no relevant content about the NaCl (salt stress) treatment, however, the results were presented in figure 6.
10. Lines 162-163: Why root activity was higher in control group than bacillus treatment in the third week is not mentioned? Can authors please discuss this result in discussion?
11. Lines 165-166: Similar problem with total phosphorus content in the second week. I am wondering if the phosphorus content can be related to fertilizer management, can authors provide detailed fertilization record in materials and methods?
12. Line 246: I didn’t find authors measured stem dry weight results.
13. Lines 246-247: According to Figure 1, there is a significant difference on root fresh weight between bacillus treatment and control.
14. The discussion part needs to be improved extensively. Right now, each paragraph looks like a brief introduction from reference plus a summary of results. Potential questions raised in results, for example, in my comments 10&11 were not discussed. The relationship between the citation and results of this study was not discussed.
15. In figure 5, do authors mean the total fresh weight of quinoa seedlings, or shoot, or root fresh weight?
16. The statement in conclusion is not precise. For example, bacillus decreased the root activity after 7 days of salt treatment instead of improving. Similar problems can be found in many other result summarizations in conclusions, please rewrite the conclusions and related parts in abstract.

·

Basic reporting

Review Report:
In this paper, Author has demonstrated the “Improved salt tolerance of quinoa by Bacillus amyloliquefaciens 11B91”. The work is interesting, but following queries must be resolved before it publishing in the journal.

1. Botanical name of quinoa add in title.
2. Title should need modification like Effect of Bacillus amyloliquefaciens 11B91on the growth of quinoa under salt tolerance or Enhancement the growth of Quinoa by Bacillus amyloliquefaciens 11B91 under salt tolerance , It should be informative and interesting for readers. This is only suggestions.
3. Abstract was not written well. Add the organism name with strain no. and mention the salt percentage . Authors give significant data in abstract.
4. Line 7: plant rhizosphere growth-promoting bacteria, it will be Plant growth-promoting rhizobacteria.
5. Introduction: Recent literature and references should add in introduction part in support of study.
6. Line 69-71: rewrite
7. In methodology, The authors should mention how much inoculation were used, give full
protocol with reference.
8. Line 79: LB liquid medium, mention the full form with composition of media and add reference also.
9. Mention the layout of the treatment used in the MSS with salt and without salt.
10. Write the name of the organism along with the strain number throughout the MSS .
11. In All fig- statistically significant different between treatments should be mention with different alphabet like a, b, c, ab, cd……..
12. . In fig- Statistically significant is not properly displayed.
13. References used in this study are very old, mostly -2001-2008, which should be updated
14. I suggest the authors to cite the following relevant and recent references
 Frontiers in Plant Science, 13:952212. doi: https://dx.doi.org/10.3389/fpls.2022.952212,
 Sustainability. 14,490. https://doi.org/10.3390/su14010490,
 Microorganisms, 9, 1491. https://doi.org/10.3390/microorganisms9071491.
 International Journal of Life Sciences and Applied Sciences, 2(2): 27-36, ISSN (online):
2582-5178,
 J. Indian bot. Soc. 100(1&2):30-41,
 Physiology and Molecular Biology of Plants. 26, pages1847–1854. doi: https://dx.doi.org/10.1007/s12298-020-00852-9,
 Indian Journal of Experimental Biology. 58, 115-121
 VEGETOS: An International Journal of Plant Research & Biotechnology: 31(2):
130-135, Doi: https://dx.doi.org/10.5958/2229-4473.2018.00064.2,
 International Journal of Agriculture, Environment and Biotechnology: 11(1): 01- 13,
 International Journal of current Microbiology and applied sciences: 6(12): 3500-3518.
doi: https://doi.org/10.20546/ijcmas.2017.612.408,

Experimental design

Mention the layout of the treatment used in the MSS with salt and without salt.

Validity of the findings

In All fig- statistically significant different between treatments should be mention with different alphabet like a, b, c, ab, cd……..
12. . In fig- Statistically significant is not properly displayed.

Additional comments

References used in this study are very old, mostly -2001-2008, which should be updated

---

## Round 0.2 · Minor Revisions

1) The title may be changed to Bacillus amyloliquefaciens 11B91 inoculation enhances the growth of Quinoa (Chenopodium quinoa Willd.) under salt stress
2) Line 1 - quinoa - The first letter should be a capital letter
3) There are still a few typos and grammar errors
4) Line .. The aim of this study was to explore.... It should proceed with the line "In this paper, the growth-promoting effect ...."
5) Line The aim of this study was to explore. Should be rephrased
60
5) Recent references (2021-2023) on PGPR in salt stress are still missing.

Reviewer 1 ·

Basic reporting

I appreciate the authors' careful revision. My comments have been fully addressed, and I have no further comments. I recommend that this manuscript be accepted for publication.

Experimental design

no comment

Validity of the findings

no comment

Additional comments

no comment

---

## Round 0.3 · accepted · Accept

Thank you for revising the manuscript as per the comments and suggestions of the reviewers.

Reviewer 1 ·

Basic reporting

I recommend that this manuscript be accepted for publication.

Experimental design

no comment

Validity of the findings

no comment